

# Exact physical quantities of a competing spin chain in the thermodynamic limit

Pengcheng Lu[1,2], Yi Qiao[1,2*], Junpeng Cao[2,3,4,5†] and Wen-Li Yang[1,5,6‡]

**1** Institute of Modern Physics, Northwest University, Xian 710127, China
**2** Beijing National Laboratory for Condensed Matter Physics, Institute of Physics, Chinese Academy of Sciences, Beijing 100190, China
**3** School of Physical Sciences, University of Chinese Academy of Sciences, Beijing 100049, China
**4** Songshan Lake Materials Laboratory, Dongguan, Guangdong 523808, China
**5** Peng Huanwu Center for Fundamental Theory, Xian 710127, China
**6** Shaanxi Key Laboratory for Theoretical Physics Frontiers, Xian 710127, China

★ qiaoyi_joy@foxmail.com , † junpengcao@iphy.ac.cn ,
‡ wlyang@nwu.edu.cn

## Abstract

We study the exact physical quantities of a competing spin chain which contains many interesting and meaningful couplings including the nearest neighbor, next nearest neighbor, chiral three spins, Dzyloshinsky-Moriya interactions and unparallel boundary magnetic fields in the thermodynamic limit. We obtain the density of zero roots, surface energies and elementary excitations in different regimes of model parameters. Due to the competition of various interactions, the surface energy and excited spectrum show many different pictures from those of the Heisenberg spin chain.


## 1  Introduction

Quantum integrable models [1] are very important to analyze some non-pertubative properties of quantum field/string theory [2,3]. Moreover, the exact solutions and physical properties of these models can provide the strict benchmarks for many important physics issues, and sometimes it can exactly predict and explain the results of experiments [4–6]. In recent years, the study of quantum integrable models play an important role in the non-equilibrium statistical physics [7–10], condensed matter physics [11], cold atom physic [12,13], superstring theory AdS/CFT [14–16] and so on.

For the integrable models with $U(1)$ symmetry, the exact solutions of the models can be obtained by the conventional Bethe ansatz. In addition, due to the homogeneous Bethe ansatz equations (BAEs) and the regular pattern of the Bethe roots, the thermodynamic properties can be directly calculated by the thermodynamic Bethe ansatz (TBA) [17,18]. When the $U(1)$ symmetry of integrable systems is broken, the off-diagonal Bethe ansatz can be used to solve the systems based on the algebraic analysis [19]. However, since the exact solutions of the systems are described by the inhomogeneous $T-Q$ relations [20,21] and the resulting inhomogeneous BAEs have the inhomogeneous term, the pattern of Bethe roots is not clear and the TBA method can not be applied. Recently, a novel Bethe ansatz scheme has been proposed to calculate the physical quantities of quantum integrable systems with or without $U(1)$ symmetry [22,23]. The key point of the scheme lies in parameterizing the eigenvalue of transfer matrix by its zero roots instead of the Bethe roots. Through this method, the homogeneous BAEs and the well-defined patterns of zero roots can be obtained. Based on them, the thermodynamic properties and exact physical quantities of the systems in the thermodynamic limit can also be calculated. In this paper, we study an isotropic quantum spin chain which includes the nearest neighbor (NN) [24], next nearest neighbor (NNN) [25], Dzyloshinsky-Moriya (DM) interactions [26,27], chirality three-spin couplings [28] and unparallel boundary magnetic fields [29]. The density of zero roots, surface energy and elementary excitations in different regimes of model parameters are obtained.

The paper is organized as follows. Section 2 serves as an introduction to the model and explain its integrability. In section 3, we give the patterns of zero roots in the different regimes of model parameters. In section 4, we calculate the surface energies induced by the boundary magnetic fields. In section 5, we study the typical bulk elementary excitations in the system. The boundary excitations are computed in section 6. In section 7, we calculate the surface energies in ferromagnetic regime. Concluding remarks are given in section 8. A simple method is introduced in Appendix A..

## 2  Integrability of the model

The model Hamiltonian reads

$$H = H_{bulk} + H_L + H_R. \tag{1}$$

Here $H_{bulk}$ describe the interactions in the bulk which includes the NN, NNN and chiral three spin couplings with the form of

$$H_{bulk} = \sum_{j=1}^{2N-1} \left\{ J_1 \vec{\sigma}_j \cdot \vec{\sigma}_{j+1} + J_2 \vec{\sigma}_j \cdot \vec{\sigma}_{j+2} + J_3 (-1)^j \vec{\sigma}_{j+1} \cdot (\vec{\sigma}_j \times \vec{\sigma}_{j+2}) \right\}, \tag{2}$$

where $\sigma_j^\alpha (\alpha = x, y, z)$ is the Pauli matrix along the $\alpha$-direction on the $j$-th site, and $2N$ is the number of sites. We note that the convention $\vec{\sigma}_{2N+1} = 0$ has been used. $H_L$ quantifies the left boundary terms which includes the boundary magnetic field along the $z$-direction and the anisotropic and DM interactions of the first bond

$$H_L = \frac{1 - 4a^2}{p^2 - a^2} [p\sigma_1^z - a^2 \sigma_1^z \sigma_2^z - iap D_1^z \cdot (\vec{\sigma}_1 \times \vec{\sigma}_2)], \tag{3}$$

where $p$ is the strength of magnetic field, $a^2$ and $ap$ quantify the spin-exchanging and DM interactions respectively, and $D_1^z$ is the unit vector along the $z$-direction. $H_R$ characterizes the right boundary terms which includes the boundary magnetic field lies in the $x-z$ plane, anisotropic and DM interactions of the last bond also constrained in the $x-z$ plane. Thus $H_R$ reads

$$H_R = \frac{4a^2 - 1}{a^2\xi^2 + a^2 - q^2} \Big[ q(\xi\sigma_{2N}^x + \sigma_{2N}^z) - a^2(\xi\sigma_{2N-1}^x + \sigma_{2N-1}^z)(\xi\sigma_{2N}^x + \sigma_{2N}^z)$$
$$- iaq(\xi D_{2N}^x + D_{2N}^z) \cdot (\vec{\sigma}_{2N} \times \vec{\sigma}_{2N-1}) \Big], \tag{4}$$

where $q$ and $\xi$ are the boundary parameters, $D_{2N}^x$ is the unit vector along the $x$-direction and $D_{2N}^z$ is the unit vector along the $z$-direction. We should note that the boundary fields are unparallel boundary and the $U(1)$ symmetry of the system are broken. The hermitian of the Hamiltonian (1) requires that the model parameter $a$ is pure imaginary and the boundary parameters $p$, $q$, $\xi$ are real. Moreover, the integrability of the system (1) requires that the couplings $J_1, J_2, J_3$ satisfy the relationships

$$J_1 = 1 + c_j(\delta_{j,1} + \delta_{j,2N-1}), \qquad J_2 = -2a^2, \qquad J_3 = ia, \tag{5}$$

$$c_1 = \frac{a^2(1 - 2a^2 - 2p^2)}{p^2 - a^2}, \qquad c_{2N-1} = 2a^2 + \frac{a^2(4q^2 - \xi^2 - 1)}{a^2\xi^2 + a^2 - q^2}, \tag{6}$$

where the index $j$ is the summation index in $H_{bulk}$ (2). The Hamiltonian (1) is constructed by using the $R$-matrix and the reflection matrices $K^\pm$ based on the quantum inverse scattering method. The $R$-matrix defined in the tensor space $V_1 \otimes V_2$ is

$$R_{1,2}(u) = u + P_{1,2} = u + \frac{1}{2}(1 + \vec{\sigma}_1 \cdot \vec{\sigma}_2), \tag{7}$$

where $u$ is the spectral parameter and $P_{1,2}$ is the permutation operator. The $R$-matrix (7) satisfies the quantum Yang-Baxter equation (QYBE),

$$R_{1,2}(u_1 - u_2)R_{1,3}(u_1 - u_3)R_{2,3}(u_2 - u_3) = R_{2,3}(u_2 - u_3)R_{1,3}(u_1 - u_3)R_{1,2}(u_1 - u_2). \tag{8}$$

The reflection matrix $K_1^-(u)$ defined the space $V_1$ is

$$K_1^-(u) = \begin{pmatrix} p+u & \\ & p-u \end{pmatrix}, \tag{9}$$

which satisfies the reflection equation (RE)

$$R_{1,2}(\lambda - u)K_1^-(\lambda)R_{2,1}(\lambda + u)K_2^-(u) = K_2^-(u)R_{1,2}(\lambda + u)K_1^-(\lambda)R_{2,1}(\lambda - u), \tag{10}$$

where $R_{2,1}(u) = P_{1,2}R_{1,2}(u)P_{1,2}$. The dual reflection matrix $K_1^+(u)$ is

$$K_1^+(u) = \begin{pmatrix} q+u+1 & \xi(u+1) \\ \xi(u+1) & q-u-1 \end{pmatrix}, \tag{11}$$

satisfying the dual reflection equation

$$R_{1,2}(-\lambda+u)K_1^+(\lambda)R_{2,1}(-\lambda-u-2)K_2^+(u) = K_2^+(u)R_{1,2}(-\lambda-u-2)K_1^+(\lambda)R_{2,1}(-\lambda+u). \tag{12}$$

The monodromy matrix $T_0(u)$ and the reflecting one $\widehat{T}_0(u)$ are constructed by the $R$-matrices as

$$T_0(u) = R_{0,2N}(u+a+\theta_{2N})R_{0,2N-1}(u-a-\theta_{2N-1})\cdots R_{0,2}(u+a+\theta_2)R_{0,1}(u-a-\theta_1),$$
$$\widehat{T}_0(u) = R_{0,1}(u+a+\theta_1)R_{0,2}(u-a-\theta_2)\cdots R_{0,2N-1}(u+a+\theta_{2N-1})R_{0,2N}(u-a-\theta_{2N}), \tag{13}$$

where $V_0$ is the auxiliary space, $\otimes_{j=1}^{2N}V_j$ is the quantum space, and $\{\theta_j | j = 1, \cdots, 2N\}$ are the inhomogeneity parameters. The transfer matrix $t(u)$ is defined as

$$t(u) = tr_0 \left\{ K_0^+(u)T_0(u)K_0^-(u)\widehat{T}_0(u) \right\}, \tag{14}$$

where $tr_0$ means the partial trace over the auxiliary space. The Hamiltonian (1) is generated by the transfer matrix as

$$H = -\frac{1}{2}(4a^2-1)\left(\frac{\partial \ln t(u)}{\partial u}\Big|_{u=a} + \frac{\partial \ln t(u)}{\partial u}\Big|_{u=-a}\right)\Big|_{\{\theta_j\}=0} - c_0, \tag{15}$$

where

$$c_0 = -(2N-1)(2a^2-1) - \frac{2a^4-6a^2+1}{a^2-1},$$
$$c_2 = 8(1-4a^2)^{2N-2}(p^2-a^2)(a^2-1)(a^2\xi^2+a^2-q^2). \tag{16}$$

The QYBE (8), the RE (10) and its dual (12) guarantee the integrability of the model described by the Hamiltonian given by (1). Moreover, using the properties of the $R$-matrix one may easily prove that $t(u) = t(-u-1)$ and the following operator identities [19]

$$t(\theta_j+a)t(\theta_j+a-1) = a(\theta_j+a)d(\theta_j+a-1), \qquad j = 1, \cdots, 2N, \tag{17}$$

where

$$a(u) = \frac{2u+2}{2u+1}(u+p)\left[(1+\xi^2)^{\frac{1}{2}}u+q\right]\prod_{j=1}^{2N}(u+\theta_j+a+1)(u-\theta_j-a+1), \tag{18}$$
$$d(u) = a(-u-1).$$

From the definition (14), we know that the transfer matrix $t(u)$ is a polynomial operator of $u$ with the degree $4N+2$. Denote the eigenvalue of the transfer matrix $t(u)$ as $\Lambda(u)$. From above analysis, we know that the eigenvalue $\Lambda(u)$ satisfies

$$\Lambda(u) = \Lambda(-u-1), \tag{19}$$

$$\Lambda(u) = 2u^{4N+2} + \cdots, \quad u \to \pm\infty, \tag{20}$$

$$\Lambda(0) = 2pq\prod_{j=1}^{2N}(1-\theta_j-a)(1+\theta_j+a) = \Lambda(-1), \tag{21}$$

$$\Lambda(\theta_j+a)\Lambda(\theta_j+a-1) = a(\theta_j+a)d(\theta_j+a-1), \quad j = 1, \cdots, 2N. \tag{22}$$

Obviously, $\Lambda(u)$ is a degree $4N + 2$ polynomial of $u$ and can be parameterized as

$$\Lambda(u) = 2 \prod_{j=1}^{2N+1} \left( u - z_j + \frac{1}{2} \right)\left( u + z_j + \frac{1}{2} \right), \tag{23}$$

where $\{z_j | j = 1, \cdots, 2N + 1\}$ are the zero roots of the polynomial. Putting the parameterizing (23) into (22), we obtain the BAEs

$$4 \prod_{l=1}^{2N+1} \left( \theta_j + a - z_l + \frac{1}{2} \right)\left( \theta_j + a + z_l + \frac{1}{2} \right)\left( \theta_j + a - z_l - \frac{1}{2} \right)\left( \theta_j + a + z_l - \frac{1}{2} \right)$$
$$= a(\theta_j + a)d(\theta_j + a - 1), \qquad j = 1, \cdots, 2N. \tag{24}$$

The above $2N$ equations and (21) can determine the $2N + 1$ unknowns $\{z_j\}$ completely. In the homogeneous limit $\{\theta_j = 0 | j = 1, \cdots, 2N\}$, Eq. (21) is replaced by

$$\Lambda(0) = 2 p q (1 - a^2)^{2N}, \tag{25}$$

and Eq. (22) becomes

$$[\Lambda(u + a)\Lambda(u + a - 1)]^{(n)}|_{u=0} = [a(u + a)d(u + a - 1)]^{(n)}|_{u=0}, \tag{26}$$

where the superscript $(n)$ indicates the $n$-th order derivative and $n = 0, 1, \cdots, 2N - 1$. Eqs. (25) and (26) can determine the $2N + 1$ zeros roots $\{z_j\}$ in the homogeneous limit in finite system size. Moreover, the energy spectrum of the Hamiltonian (1) can be determined by the zero roots as

$$E = -\pi(4a^2 - 1) \sum_{j=1}^{2N+1} [a_1(iz_j - ia) + a_1(iz_j + ia)] - c_0, \tag{27}$$

where the function $a_n(u)$ is given by

$$a_n(u) = \frac{1}{2\pi} \frac{n}{u^2 + n^2/4}. \tag{28}$$

By solving the BAEs Eqs. (25) and (26), we can obtain all the eigen-energies of the system (1).

## 3 Patterns of zero roots

We first study the solutions of zero roots $\{z_j\}$ at the ground state. For convenient, we choose all the inhomogeneity parameters to be imaginary, $\{\theta_j \equiv i\bar{\theta}_j\}$, and let $\{\bar{z}_j \equiv -iz_j\}$. In addition, we set the boundary parameters as $p > 0$ and $\bar{q} = q(1 + \xi^2)^{-\frac{1}{2}}$. From the numerical calculation and algebraic analysis, we find that the distribution of the $\bar{z}$-roots at the ground state can be divided into following six different regimes in the upper $p - \bar{q}$ plane, as shown in Fig. 1.

1) In the regime I, where $0 \le p < \frac{1}{2}, 0 \le \bar{q} < \frac{1}{2}$, all the $\bar{z}$-roots form $2N - 2$ conjugate pairs as $\{\bar{z}_j \sim \tilde{z}_j \pm i | j = 1, \cdots, 2N - 2\}$ with real $\{\tilde{z}_j\}$, two boundary conjugate pairs $\{\pm i(|p| + \frac{1}{2}), \pm i(|\bar{q}| + \frac{1}{2})\}$ and two symmetrical real roots $\bar{z}_\pm = \pm \alpha$. The numerical check with $2N = 8$ is shown in Fig. 2a. In the thermodynamic limit, two symmetrical real roots $\pm \alpha$ would tend to infinity and contribute nothing to the ground state energy. These two real roots correspond to the Majorana modes at the two boundaries.

2) In the regime II, where $0 \le p < \frac{1}{2}, -\frac{1}{2} \le \bar{q} < 0$, as shown in Fig. 2b, all the $\bar{z}$-roots form $2N - 2$ conjugate pairs, two boundary conjugate pairs $\{\pm i(|p| + \frac{1}{2}), \pm i(|\bar{q}| + \frac{1}{2})\}$ and one pure imaginary conjugate pair $\pm i\beta$ with $\beta > \min(|p|, |\bar{q}|)$.

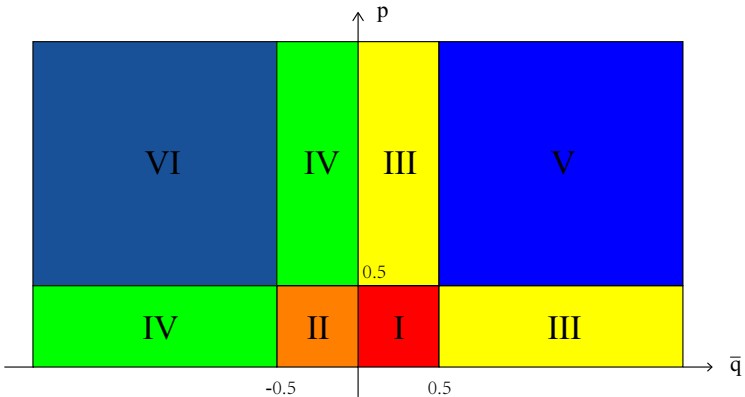

Figure 1: The distribution of $\bar{z}$-roots at the ground state in the upper $p - \bar{q}$ plane.

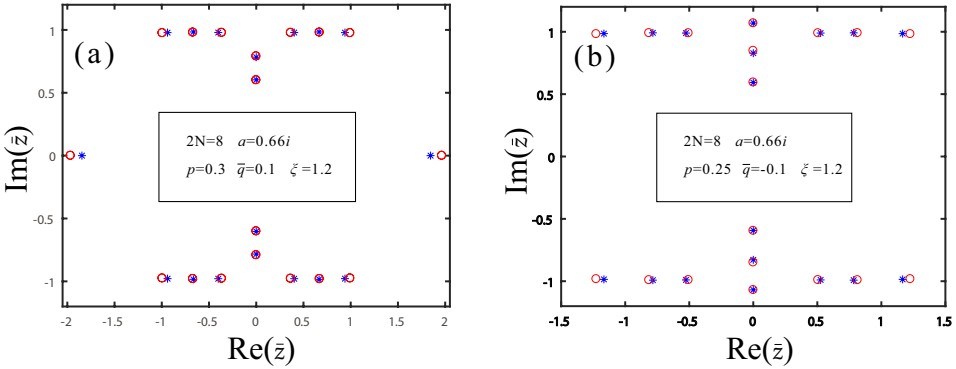

Figure 2: Pattern of $\bar{z}$-roots at the ground state in regimes I (a) and II (b) with $2N = 8$. The blue asterisks indicate $\bar{z}$-roots for $\{\bar{\theta}_j = 0 | j = 1, \cdots, 2N\}$ and the red circles specify $\bar{z}$-roots with the inhomogeneity parameters $\{\bar{\theta}_j = 0.1(j - N - 0.5) | j = 1, \cdots, 2N\}$.

3) In the regime III, where $p \geq \frac{1}{2}, 0 \leq \bar{q} < \frac{1}{2}$ or $0 \leq p < \frac{1}{2}, \bar{q} \geq \frac{1}{2}$, as shown in Fig. 3a, all the $\bar{z}$-roots form $2N - 2$ conjugate pairs, one boundary conjugate pair $\pm i[\min(|p|, |\bar{q}|) + \frac{1}{2}]$, two symmetrical real roots $\bar{z}_{\pm} = \pm \alpha$, and one pure imaginary conjugate pair $\pm i\beta$ with $\beta > \min(|p|, |\bar{q}|)$.

4) In the regime IV, where $p \geq \frac{1}{2}, -\frac{1}{2} \leq \bar{q} < 0$ or $0 \leq p < \frac{1}{2}, \bar{q} \leq -\frac{1}{2}$, as shown in Fig. 3b, all the $\bar{z}$-roots form $2N$ conjugate pairs and one boundary conjugate pair $\pm i[\min(|p|, |\bar{q}|) + \frac{1}{2}]$.

5) In the regime V, where $p \geq \frac{1}{2}, \bar{q} \geq \frac{1}{2}$, as shown in Fig. 3c, all the $\bar{z}$-roots form $2N$ conjugate pairs and two symmetrical real roots $\bar{z}_{\pm} = \pm \alpha$.

6) In the regime VI, where $p \geq \frac{1}{2}, \bar{q} \leq -\frac{1}{2}$, as shown in Fig. 3d, all the $\bar{z}$-roots form $2N$ conjugate pairs and one pure imaginary conjugate pair $\pm i\beta$ with $\beta > \min(|p|, |\bar{q}|)$.

We also find that the choice of pure imaginary inhomogeneities $\{\bar{\theta}_j\}$ does not change the patterns of zero roots $\{\bar{z}_j\}$ but the roots density, as shown in Fig. 2. This result allows us to calculate the physical quantities such as the surface energy and the elementary excitations of the system in the thermodynamic limit with the help of suitable $\{\bar{\theta}_j\}$ [23].

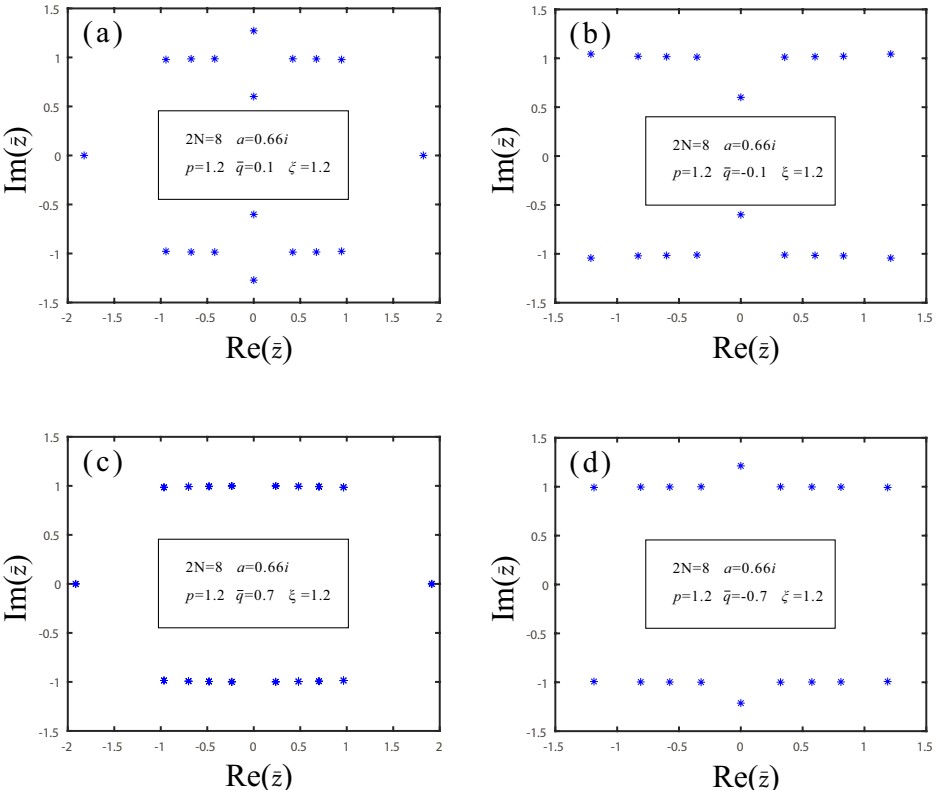

Figure 3: $(a)$-$(d)$ Patterns of $\bar{z}$-roots for $\{\bar{\theta}_j = 0 | j = 1, \cdots, 2N\}$ at the ground state in regimes III-VI with $2N = 8$.

# 4 Surface energy

Now, we consider the surface energy induced by the boundaries. The surface energy is defined by $E_b = E_g - E_p$, where $E_g$ is the ground state energy of present system and $E_p$ is the ground state energy of the corresponding periodic chain. In the thermodynamic limit, the distribution of $\tilde{z}$-roots can be characterized by the density $\rho(\tilde{z})$. Furthermore, we assume that the density of inhomogeneity parameters $1/[2N(\bar{\theta}_j - \bar{\theta}_{j-1})]$ has the continuum limit $\sigma(\bar{\theta})$.

In regime I, substituting the corresponding pattern of $\bar{z}$-roots into BAEs (24) and taking the logarithm of the absolute value, we have

$$
\begin{aligned}
&\ln|4| + \sum_{l=1}^{2N-1}\left[\ln\left|\bar{\theta}_j + \bar{a} - \tilde{z}_l + \frac{3i}{2}\right| + \ln\left|\bar{\theta}_j + \bar{a} - \tilde{z}_l + \frac{i}{2}\right| + \ln\left|\bar{\theta}_j + \bar{a} - \tilde{z}_l - \frac{i}{2}\right| + \ln\left|\bar{\theta}_j + \bar{a} - \tilde{z}_l - \frac{3i}{2}\right|\right] \\
&\quad + \ln\left|\left(\bar{\theta}_j + \bar{a} - \alpha + \frac{i}{2}\right)\left(\bar{\theta}_j + \bar{a} - \alpha - \frac{i}{2}\right)\right| + \ln\left|\left(\bar{\theta}_j + \bar{a} + \alpha + \frac{i}{2}\right)\left(\bar{\theta}_j + \bar{a} + \alpha - \frac{i}{2}\right)\right| \\
&\quad + \ln|(\bar{\theta}_j + \bar{a} - i|p|)(\bar{\theta}_j + \bar{a} + i|p|)| + \ln|(\bar{\theta}_j + \bar{a} - i|p| - i)(\bar{\theta}_j + \bar{a} + i|p| + i)| \\
&\quad + \ln|(\bar{\theta}_j + \bar{a} - i|\bar{q}|)(\bar{\theta}_j + \bar{a} + i|\bar{q}|)| + \ln|(\bar{\theta}_j + \bar{a} - i|\bar{q}| - i)(\bar{\theta}_j + \bar{a} + i|\bar{q}| + i)| \\
&= \ln|(\bar{\theta}_j + \bar{a} + i)(\bar{\theta}_j + \bar{a} - i)| - \ln\left|\left((\bar{\theta}_j + \bar{a}) + \frac{i}{2}\right)\left((\bar{\theta}_j + \bar{a}) - \frac{i}{2}\right)\right| \\
&\quad + \ln|(\bar{\theta}_j + \bar{a} + ip)(\bar{\theta}_j + \bar{a} - ip)| + \ln|((1 + \xi^2)^{\frac{1}{2}}(\bar{\theta}_j + \bar{a}) + iq)((1 + \xi^2)^{\frac{1}{2}}(\bar{\theta}_j + \bar{a}) - iq)| \\
&\quad + \sum_{k=1}^{2N}[(\ln|(\bar{\theta}_j - \bar{\theta}_k + i)(\bar{\theta}_j - \bar{\theta}_k - i)| + \ln|(\bar{\theta}_j - \bar{\theta}_k + 2\bar{a} + i)(\bar{\theta}_j - \bar{\theta}_k + 2\bar{a} - i)|], \quad (29)
\end{aligned}
$$

where $\bar{a} = -ia$. In the thermodynamic limit, we assume that the zero roots and inhomogeneities have continuum densities

$$\rho(\tilde{z}) = \frac{1}{2N(\tilde{z}_{j+1} - \tilde{z}_j)}, \quad \sigma(\bar{\theta}) = \frac{1}{2N(\bar{\theta}_{j+1} - \bar{\theta}_j)} \, .$$

Taking the continuum limit of Eq. (29) and replacing $\bar{\theta}_j$ with $\lambda$, we obtain

$$2N \int_{-\infty}^{\infty} [b_1(\lambda + \bar{a} - \tilde{z}) + b_3(\lambda + \bar{a} - \tilde{z})] \rho(\tilde{z}) d\tilde{z} + b_1(\lambda + \bar{a} + \alpha) + b_1(\lambda + \bar{a} - \alpha)$$

$$= 2N \int_{-\infty}^{\infty} [b_2(\lambda - \bar{\theta}) + b_2(\lambda + \bar{\theta} + 2\bar{a})] \sigma(\bar{\theta}) d\bar{\theta} + b_2(\lambda + \bar{a}) - b_1(\lambda + \bar{a})$$

$$- b_{2|p|+2}(\lambda + \bar{a}) - b_{2|\bar{q}|+2}(\lambda + \bar{a}), \tag{30}$$

where $b_n(\lambda) = \frac{1}{2\pi} \frac{2\lambda}{\lambda^2 + n^2/4}$. Eq.(30) is a convolution equation and can be solved by the Fourier transformation. The solution of $\tilde{z}$-roots density is

$$\tilde{\rho}(k) = [4N\tilde{b}_2(k) \cos(\bar{a}k)\tilde{\sigma}(k) + \tilde{b}_2(k) - \tilde{b}_1(k) - \tilde{b}_{2|p|+2}(k)$$

$$- \tilde{b}_{2|\bar{q}|+2}(k) - 2\tilde{b}_1(k) \cos(\alpha k)] / [2N(\tilde{b}_1(k) + \tilde{b}_3(k))], \tag{31}$$

where $\tilde{b}_n(k) = sign(k)ie^{-|nk|}$. From now on, we use $\sigma(\theta) = \delta(\theta)$. In the thermodynamic limit, $\alpha$ tends to infinity. The ground state energy of the Hamiltonian (1) can thus be expressed as

$$E_{g1} = N(4a^2 - 1) \int_{-\infty}^{\infty} [\tilde{a}_1(k) - \tilde{a}_3(k)] \cos(\bar{a}k)\tilde{\rho}(k) dk - c_0$$

$$- (4a^2 - 1) \left[ \frac{|p|}{a^2 - p^2} - \frac{|p| + 1}{a^2 - (|p| + 1)^2} + \frac{|\bar{q}|}{a^2 - \bar{q}^2} - \frac{|\bar{q}| + 1}{a^2 - (|\bar{q}| + 1)^2} \right], \tag{32}$$

where $\tilde{a}_n(k) = e^{-|nk|}$ is the Fourier transformation of $a_n(\lambda)$. The ground state energy of the system with periodic boundary condition can be obtained similarly. After tedious calculation, we obtain the surface energy in the regime I as

$$E_{b1} = e_b(p) + e_b(q) + e_{b0}, \tag{33}$$

$$e_b(p) = \frac{(4a^2 - 1)}{4} \int_{-\infty}^{\infty} (1 - e^{-|k|}) \cosh(ak) \frac{e^{-|pk|}}{e^{-|k|/2} \cosh(k/2)} dk, \tag{34}$$

$$e_b(q) = \frac{(4a^2 - 1)}{4} \int_{-\infty}^{\infty} (1 - e^{-|k|}) \cosh(ak) \frac{e^{-|(q/\sqrt{1+\xi^2})k|}}{e^{-|k|/2} \cosh(k/2)} dk, \tag{35}$$

$$e_{b0} = \frac{(4a^2 - 1)}{4} \int_{-\infty}^{\infty} (1 - e^{-|k|}) \cosh(ak) \frac{e^{-|k|} - e^{-|k|/2}}{e^{-|k|/2} \cosh(k/2)} dk. \tag{36}$$

From Eq.(33), we see that the surface energy $E_{b1}$ can be divided into three terms. $e_b(p)$ and $e_b(q)$ are the contributions of left and right boundaries, respectively. $e_{b0}$ exactly equals to the surface energy induced by the free boundaries.

In the regime II, taking the logarithm then the derivative of the absolute value of BAE (24), we have

$$
2N \int_{-\infty}^{\infty} [b_1(\lambda + \bar{a} - \tilde{z}) + b_3(\lambda + \bar{a} - \tilde{z})]\rho(\tilde{z})d\tilde{z}
$$
$$
= 2N \int_{-\infty}^{\infty} [b_2(\lambda - \bar{\theta}) + b_2(\lambda + \bar{\theta} + 2\bar{a})]\sigma(\bar{\theta})d\bar{\theta} + b_2(\lambda + \bar{a}) - b_1(\lambda + \bar{a})
$$
$$
- b_{2|p|+2}(\lambda + \bar{a}) - b_{2|\bar{q}|+2}(\lambda + \bar{a}) - b_{2|\beta|+1}(\lambda + \bar{a}) - b_{2|\beta|-1}(\lambda + \bar{a}). \tag{37}
$$

The Fourier transform gives

$$
\tilde{\rho}(k) = [4N\tilde{b}_2(k)\cos(\bar{a}k)\tilde{\sigma}(k) + \tilde{b}_2(k) - \tilde{b}_1(k) - \tilde{b}_{2|p|+2}(k) - \tilde{b}_{2|\bar{q}|+2}(k)
$$
$$
- \tilde{b}_{2|\beta|+1}(k) - \tilde{b}_{2|\beta|-1}(k)]/[2N(\tilde{b}_1(k) + \tilde{b}_3(k))]. \tag{38}
$$

Then we obtain the surface energy in this regime as

$$
E_{b2} = e_b(p) + e_b(q) + e_{b0}, \tag{39}
$$

where $e_b(p)$, $e_b(q)$ and $e_{b0}$ are given by Eqs.(34)-(36), respectively. It is clear that the forms of surface energies in the regimes I and II are the same, although the resulted values are different.

We further calculate the surface energies in the rest regimes and the result is that all the surface energies can be expressed as the form of Eq.(33). The reason is that the bare contributions of the boundary conjugate pairs to the ground state energy are exactly canceled by those of the back flow of continuum root density, as happened in the diagonal open boundary case.

The surface energies $E_b$ with certain $a$ versus the different values of boundary parameter $p$ are shown in Fig. 4(a). If $a = 0$, all the NNN, chiral three spin and DM interactions are zero and the system (1) degenerates into the Heisenberg spin chain with unparallel boundary fields. From the blue dotted lines in Fig. 4(a), we see that the surface energy of Heisenberg spin chain is smaller than zero, and is monotonically increasing with the increasing of $|p|$. When $p = 0$, the surface energy is divergent, this is because that the strength of boundary magnetic field is quantified by $1/p$. The results are similar to the those of the Heisenberg spin chain with parallel boundary fields [30,31]. While for the present model with $a \neq 0$, the surface energies can be larger or smaller than zero, and have two peaks and three minimums at some special values of $|p|$. At the point of $p = 0$, the surface energy arrives at its minimum. The surface energy is smaller than that of Heisenberg spin chain if $|p|$ is large, and is larger than that of Heisenberg spin chain if $|p|$ is smalle.

The surface energies $e_b(p)$ with fixed $a$ versus $p$ are shown in Fig. 4(b). Comparing Figs. 4(a) and (b), we find that if $|p|$ is large which means that the boundary field is small, due to the existence of NNN, chiral three spin and DM interactions, the surface energy is smaller than that of the Heisenberg spin chain. We should note that the relation between $e_b(\bar{q})$ and $\bar{q}$ is the same as that between $e_b(p)$ and $p$, where $\bar{q} = q/\sqrt{1 + \xi^2}$.

The strength of boundary magnetic field along the $z$-direction is quantified by $p$ or $q$ up to a normalized scalar factor. The further numerical calculation of the analytical expression of surface energy shows that the curves of $E_b$ versus $q$ are similar with those of $E_b$ versus $p$. Thus we omit the figure of $E_b$ with the changing of $q$ here. In Fig. 4(c), we show the surface energies $E_b$ with given $a$ versus the boundary parameter $\xi$. The $\xi$ quantifies the twisted angle between two unparallel boundary magnetic fields, and quantifies the strength of magnetic field on the right boundary. If $\xi$ is large, the twisted angle is large. At the same time, the right boundary magnetic field is small. From the blue dotted lines in Fig. 4(c), which corresponds to the Heisenberg spin chain, we see clearly that if $\xi$ is small, the magnetic field is strong thus the induced surface energy is large, as it should be. For the present system with $a \neq 0$, if $\xi$ is

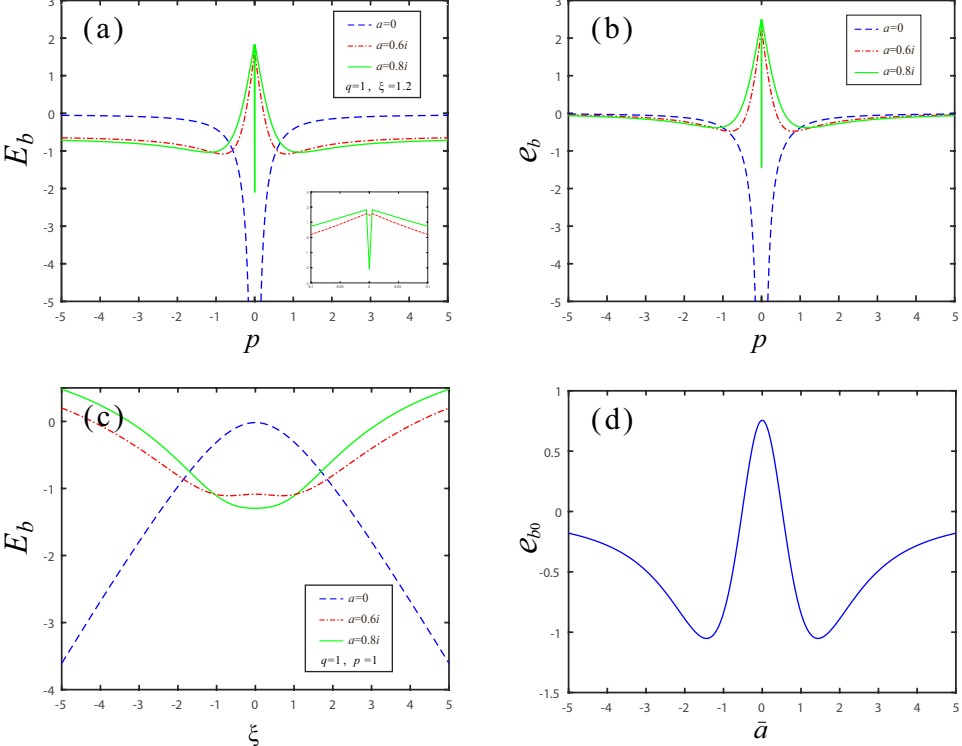

Figure 4: (a) The surface energy $E_b$ versus the boundary parameter $p$, where $a = 0, 0.6i, 0.8i$, $p = 1$ and $\xi = 1.2$. (b) The surface energy $e_b(p)$ versus the boundary parameter $p$. (c) The surface energy $E_b$ versus the boundary parameter $\xi$. (d) The surface energy $e_{b0}$ versus $a$.

small, the contributions of NNN, chiral three spin and DM interactions are large, which leads to the surface energy becomes small. Thus the behaviors of surface energies with $a = 0$ and $a \neq 0$ are totally different.

The surface energies $e_{b0}$ versus the different values of parameter $a$ are shown in Fig. 4(d). We note that the value of $e_{b0}$ at the point of $a = 0$ is the surface energy of the Heisenberg spin chain with free open boundaries.

From above explanations, we conclude that the surface energy of present system is quite different from that of the Heisenberg spin chain.

## 5 Bulk elementary excitations

Next, we study the elementary excitations in the system. We first consider the excitations in the bulk. The bulk excitations in different regimes of boundary parameters are the same. From the patterns of zero roots in the low-lying excited states, we find that the excitations can be characterized by breaking several conjugate pairs and putting the corresponding zero roots into the real axis, or the zero roots forming the conjugate pairs on the imaginary axis with more larger imaginary parts $\pm\frac{ni}{2} (n > 2)$. Thus the system has two kinds of bulk elementary excitations. The first one is quantified by four finite real roots $\{\pm\bar{z}_1, \pm\bar{z}_2\}$ and the second one is quantified by two conjugate pairs $\{\tilde{z}_n \pm \frac{ni}{2}, -\tilde{z}_n \pm \frac{ni}{2}\}$, where the distribution of rest zero roots almost does not change and the related difference between ground and excited states can be erased by the rearrangement of Fermi sea in the thermodynamic limit.

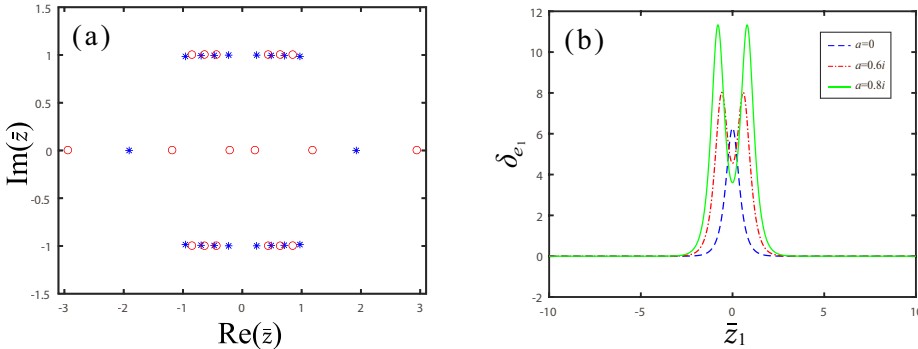

Figure 5: (a) The distribution of zero roots for $\{\bar{\theta}_j = 0 | j = 1, \cdots, 2N\}$ at the ground state (blue asterisks) and at the first kind of excited state (red circles) with $2N = 8$, $a = 0.66i$, $p = 1.2$, $\bar{q} = 0.7$ and $\xi = 1.2$. (b) The excited energies $\delta_{e_1}$ with fixed $a$ versus $\bar{z}_1$ in the thermodynamic limit.

As an example, we give the pattern of zero roots at the ground state (blue asterisks) and that at the first kind of excited sate (red circles) in the regime V with $2N = 8$, which is shown in Fig. 5a. It is clear that there are four new real roots at the excited state. In the thermodynamic limit, the density difference $\delta\tilde{\rho}_{e_1}(k)$ between the ground state and the excited state is

$$\delta\tilde{\rho}_{e_1}(k) = -\frac{\cos(\bar{z}_1 k) + \cos(\bar{z}_2 k)}{2N e^{-|k|/2} \cosh(k/2)}, \tag{40}$$

where $\bar{z}_1$ and $\bar{z}_2$ can take arbitrary continuous values in the real axis. Thus the energy carried by this kind of excitation is

$$\delta_e = \delta_{e_1}(\bar{z}_1) + \delta_{e_1}(\bar{z}_2),$$

$$\delta_{e_1}(\bar{z})|_{\bar{z}=\bar{z}_1,\bar{z}_2} = -\frac{1}{2}(4a^2 - 1)\left[\int_{-\infty}^{\infty}(1 - e^{-|k|})\cosh(ak)\cos(\bar{z}k)\cosh^{-1}(k/2)dk\right.$$

$$\left.+\frac{1}{(\bar{z}-ia)^2 + \frac{1}{4}} + \frac{1}{(\bar{z}+ia)^2 + \frac{1}{4}}\right]\Bigg|_{\bar{z}=\bar{z}_1,\bar{z}_2}$$

$$= -(4a^2 - 1)\cdot\left(\frac{\pi}{\cosh(\bar{z}+ia)} + \frac{\pi}{\cosh(\bar{z}-ia)}\right), \tag{41}$$

which covers the previous results obtained by using the conventional Bethe ansatz method for the periodic staggered ($a \neq 0$) spin chain [32]. The excited energies $\delta_{e_1}$ with given values of model parameter $a$ versus $\bar{z}_1$ are shown in Fig. 5b. From it, we see that the excited energy of the Heisenberg spin chain ($a = 0$) only has one peak at the point of $\bar{z} = 0$, while for the present model ($a \neq 0$), the excited energies have two peaks at finite $\pm\bar{z}$.

Now, we focus on the second kind of elementary excitation. In order to see the high strings ($n > 2$) excitations more clearly, we show the pattern of zero roots at the $n = 3$ excited state in Fig. 6, where the ground state is still in the regime V. In the thermodynamic limit, the density difference $\delta\tilde{\rho}_{e_n}(k)$ between the ground state and the excited state is

$$\delta\tilde{\rho}_{e_n}(k) = -\frac{(e^{-|(n+1)k|/2} + e^{-|(n-1)k|/2})\cos(\tilde{z}_n k)}{2N e^{-|k|/2}\cosh(k/2)}, \tag{42}$$

where $\tilde{z}_n$ is free. The related elementary excitation energy is

$$\delta_{e_n} = -\frac{(4a^2 - 1)}{2}\left[\int_{-\infty}^{\infty}(1 - e^{-|k|})\cosh(ak)\frac{(e^{-|(n+1)k|/2} + e^{-|(n-1)k|/2})\cos(\tilde{z}_n k)}{e^{-|k|/2}\cosh(k/2)}dk\right.$$

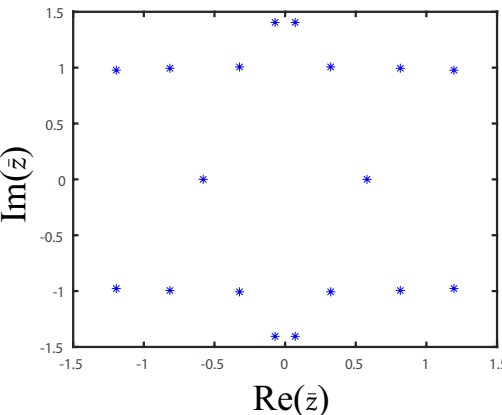

Figure 6: The distribution of $\bar{z}$-roots for $\{\bar{\theta}_j = 0 | j = 1, \cdots, 2N\}$ at the second kind of excited state with $n = 3$. Here $2N = 8$, $a = 0.66i$, $p = 1.2$, $\bar{q} = 0.7$ and $\xi = 1.2$.

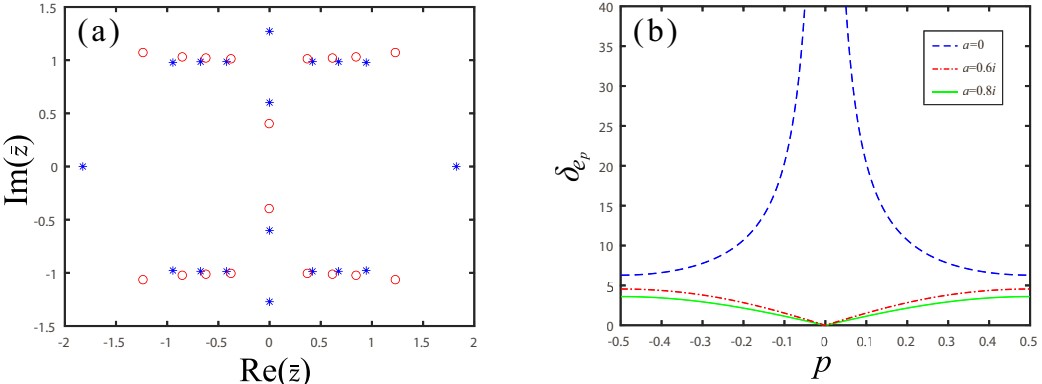

Figure 7: (a) The distribution of $\bar{z}$-roots for $\{\bar{\theta}_j = 0 | j = 1, \cdots, 2N\}$ with $2N = 8$, $a = 0.66i$, $p = 0.1$, $\bar{q} = 1.2$ and $\xi = 1.2$. Here the blue asterisks represent the pattern of zero roots at the ground state and the red circles denote those at the excited state with boundary string $i(\frac{1}{2} - |p|)$. (b) The boundary excited energy versus the boundary parameter $p$.

$$+ 2\pi(a_{n+1}(\tilde{z}_n + ia) + a_{n+1}(\tilde{z}_n - ia) - a_{n-1}(\tilde{z}_n + ia) - a_{n-1}(\tilde{z}_n - ia))]$$
$$= 0, \tag{43}$$

which indicates that the bare contributions of the conjugate pairs with $n > 2$ to the energy is exactly canceled by that of the back flow of the continuum root density. Thus the conjugate pairs with $n > 2$ contribute nothing to the energy. However, the conjugate pairs do affect the scattering matrix among the real roots [33].

## 6 Boundary elementary excitations

Next, we consider the boundary excitations. Comparing with the zero roots distributions at the ground state, we find that the boundary excitations can exist in the regimes I-IV, where the boundary parameter $-\frac{1}{2} < p < \frac{1}{2}$ or $-\frac{1}{2} < \bar{q} < \frac{1}{2}$. The typical boundary excitation is putting the boundary string from $i(|p| + \frac{1}{2})$ to $i(\frac{1}{2} - |p|)$, or from $i(|\bar{q}| + \frac{1}{2})$ to $i(\frac{1}{2} - |\bar{q}|)$. These two

new boundary strings indeed are the solutions of BAEs (24) and would appear at the low-lying excited states.

As an example, we show the pattern of zero roots at the ground state (blue asterisks) and that at the excited state (red circles) with boundary string $i(\frac{1}{2} - |p|)$ in the regime III with $2N = 8$, which is shown in Fig. 7a. We can find in the excitation, the 4 roots at $\pm\alpha$ and $\pm\beta$ of the ground state jump into the bulk string parts at $\pm i$ axes. The change of the zero roots $\pm\alpha$ and $\pm i\beta$ contribute nothing to the energy. Therefore, we omit the zero roots $\pm\alpha$ and $\pm i\beta$ in the following. The resulted density change $\delta\tilde{\rho}(k)$ between ground and excited states reads

$$\delta\tilde{\rho}_p(k) = -\frac{e^{|pk|} - e^{-|pk|}}{4N\cosh(k/2)}. \tag{44}$$

The corresponding excited energy is

$$\begin{aligned}
\delta_{e_p} &= -\frac{(4a^2-1)}{2}\left[\int_{-\infty}^{\infty}(1-e^{-|k|})\cosh(ak)\frac{\cosh(|p|k)}{e^{|k|/2}\cosh(k/2)}dk\right.\\
&\left.\quad +\frac{4|p|}{p^2-a^2}-\frac{2(|p|+a)}{(|p|+a)^2-1}-\frac{2(|p|-a)}{(|p|-a)^2-1}\right]\\
&= -\pi(4a^2-1)\cdot\Big(\csc(\pi(|p|+a))+\csc(\pi(|p|-a))\Big). \tag{45}
\end{aligned}$$

The excited energies $\delta_{e_p}$ with fixed values of $a$ versus $p$ are shown in Fig. 7b. From it, we see that the excited energy of present model is increasing with the increasing of boundary parameter $|p|$ and has a minimum at the point of $p = 0$, which is very different from that of the Heisenberg spin chain. For the latter, the excited energy is decreasing with the increase of $|p|$.

We have computed the boundary excitations in other regimes and found that the excited energies has an unified form (45), although the resulted values are different. Please note that when considering the boundary excitations in the regime of $-\frac{1}{2} < \bar{q} < \frac{1}{2}$, the $p$ in Eq.(45) should be replaced by the $\bar{q}$.

# 7 Surface energy in ferromagnetic regime

Furthermore, we study the surface energy in ferromagnetic regime. The corresponding Hamiltonian $H^{ferr}$ is the negative of Hamiltonian (1), namely

$$H^{ferr} = -H = -(H_{bulk} + H_L + H_R). \tag{46}$$

In region III ($p \geq \frac{1}{2}, 0 \leq \bar{q} < \frac{1}{2}$ or $0 \leq p < \frac{1}{2}, \bar{q} \geq \frac{1}{2}$), all the zeros $\{\bar{z}_j | j = 1, \cdots, N\}$ are real as shown in Fig. 8a. Taking the logarithm then the derivative of the absolute value of BAE (24), we have

$$\begin{aligned}
2N\int_{-\infty}^{\infty} b_1(u+\bar{a}-\tilde{z})\rho^{ferr}(\tilde{z})d\tilde{z} - b_{2|p|}(u+\bar{a}) - b_{2|\bar{q}|}(u+\bar{a})\\
= \int_{-\infty}^{\infty}[b_2(u-\bar{\theta})+b_2(u+\bar{\theta}+2\bar{a})]\sigma(\bar{\theta})d\bar{\theta} + b_2(u+\bar{a}) - b_1(u+\bar{a}). \tag{47}
\end{aligned}$$

The Fourier transform gives

$$\tilde{\rho}^{ferr}(k) = [4N\tilde{b}_2(k)\cos(\bar{a}k)\tilde{\sigma}(k)+\tilde{b}_2(k)-\tilde{b}_1(k)+\tilde{b}_{2|p|}(k)+\tilde{b}_{2|\bar{q}|}(k)]/[2N\tilde{b}_1(k)]$$

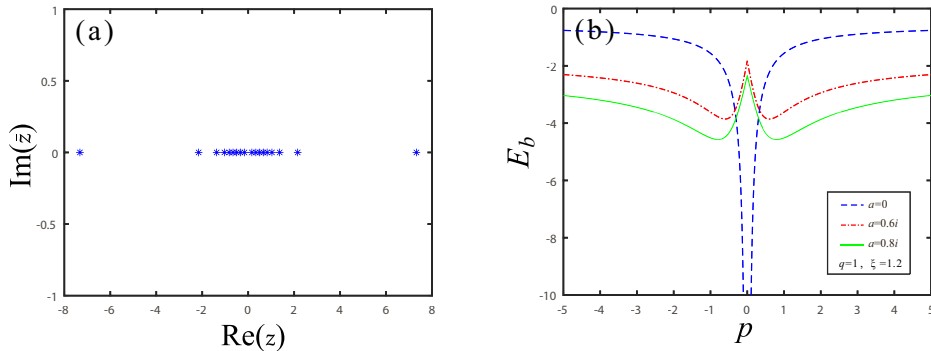

Figure 8: (a) Patterns of $\bar{z}$-roots for $\{\bar{\theta}_j = 0 | j = 1, \cdots, 2N\}$ at the ground state of the ferromagnetic case in regimes III with $2N = 8$. (b) The surface energy $E_b^{ferr}$ versus the boundary parameter $p$ in ferromagnetic case, where $a = 0, 0.6i, 0.8i$, $p = 1$ and $\xi = 1.2$.

$$= 2\tilde{a}_1(k)\cos(\bar{a}k)\tilde{\sigma}(k) + \frac{1}{2N}[\tilde{a}_1(k) - 1 + \tilde{a}_{2|p|-1}(k) + \tilde{a}_{2|\bar{q}|-1}(k)]. \qquad (48)$$

The ground state energy of the Hamiltonian (46) can thus be expressed as

$$E_g^{ferr} = N(4a^2 - 1)\int_{-\infty}^{\infty} \tilde{a}_1(k)\cos(\bar{a}k)\tilde{\rho}(k)dk + c_0$$

$$= (2N+1)(2a^2-1) - \frac{2a^4 - 6a^2 + 1}{a^2 - 1} + E_b^{ferr}, \qquad (49)$$

where the surface energy $E_b^{ferr}$ in this regime as

$$E_b^{ferr} = \frac{(4a^2-1)}{2}\int_{-\infty}^{\infty}[\tilde{a}_2(k) - \tilde{a}_1(k) + \tilde{a}_{2|p|}(k) + \tilde{a}_{2|\bar{q}|}(k)]\cos(\bar{a}k)dk$$

$$= \frac{(4a^2-1)}{2}\left[\frac{2|p|}{p^2 - a^2} + \frac{2|\bar{q}|}{\bar{q}^2 - a^2} + \frac{2}{1 - a^2} - \frac{1}{\frac{1}{4} - a^2}\right]. \qquad (50)$$

After calculation, the energy expressions in the other regions are found to be identical to Eq. (50) in region III. The surface energies $E_b^{ferr}$ with certain $a$ versus the different values of boundary parameter $p$ are shown in Fig. 8b.

## 8 Conclusions

In this paper, we have studied the exact physical quantities of a competing spin chain including the NN, NNN, chiral three-spin couplings, DM interactions and unparallel boundary magnetic fields in the thermodynamic limit. We obtained the density of zero roots, surface energy and elementary excitations in different regimes of model parameter. Due to the competition of various interactions, the excited spectrum have different behaviors from those of the isotropic Heisenberg spin chain.

# Acknowledgments

We would like to thank Prof. Y. Wang for his valuable discussions and continuous encouragement.

**Funding information** The financial supports from National Key R&D Program of China (Grant No.2021YFA1402104), the National Natural Science Foundation of China (Grant Nos. 122471 03, 12074410, 12247103, 11934015 and 11975183), Major Basic Research Program of Natural Science of Shaanxi Province (Grant Nos. 2021JCW-19 and 2017ZDJC-32), Australian Research Council (Grant No. DP 190101529), Strategic Priority Research Program of the Chinese Academy of Sciences (Grant No. XDB33000000), and the fellowship of China Postdoctoral Science Foundation (2020M680724) are gratefully acknowledged.

# A  A simple method

In the review process, one anonymous referee recommends a clear and simple method to derive the surface energy and the bulk excitations. Here, we list the referee's method. Under the simplifications that take place in the thermodynamic limit (dense distribution of zeros) one can apply techniques introduced in [34] for the excitations and in [35, 36] for the bulk properties. In the thermodynamic limit the functional relations (22) means

$$\Lambda(u)\Lambda(u-1) = a(u)d(u-1) = a(u)a(-u), \tag{A.1}$$

for all $u$ out of the physical strip. Of course this means literally for the bulk and surface terms

$$\Lambda(u) = \Lambda_{bulk}(u) \cdot \Lambda_{sur}(u), \tag{A.2}$$

$$a(u)| = a_{bulk}(u) \cdot a_{sur}(u), \qquad a_{sur}(u) := \frac{u+1}{u+\frac{1}{2}}(u+p)(u+\bar{q}), \tag{A.3}$$

that for instance

$$\Lambda_{sur}(u)\Lambda_{sur}(u-1) = a_{sur}(u)a_{sur}(-u). \tag{A.4}$$

Now introducing

$$\tilde{\Lambda}(u) := \Lambda_{sur}(-iu) \tag{A.5}$$

allows for the ansatz of a Fourier transform

$$\frac{d}{du} \log \tilde{\Lambda}(u) = \int_{-\infty}^{\infty} dk L(k)e^{iku}, \tag{A.6}$$

with a yet unknown function $L(k)$. This function can be calculated from (A.4) by taking the logarithm, the derivative and then the Fourier transform (the RHS gives an explicit function):

$$L(k)\cdot(1+e^k) = -i\cdot sign(k)\cdot(e^{-|pk|} + e^{-|\bar{q}k|} + e^{-|k|} - e^{-|k|/2}). \tag{A.7}$$

From the last equation one gets $L(k)$ and from this $\frac{d}{du} \log \tilde{\Lambda}(u)$ Fourier transform. The energy is simply obtained by

$$E_{sur} = -\frac{1}{2}(4a^2 - 1)\left( i\frac{d}{du} \log \tilde{\Lambda}(u)\Big|_{u=ia} + i\frac{d}{du} \log \tilde{\Lambda}(u)\Big|_{u=-ia} \right), \tag{A.8}$$

which straight away gives (33) of the paper.

Next, the referee derives the bulk excitations. He starts with a remark: The result (41) can be presented in a simplified, explicit form, by doing the Fourier integral resulting in:

$$\delta_{e_1}(\bar{z}) = -(4a^2 - 1) \cdot \left( \frac{\pi}{\cosh(\bar{z} + ia)} + \frac{\pi}{\cosh(\bar{z} - ia)} \right). \tag{A.9}$$

How to derive this in a most transparent manner? Define for an arbitrary excited state, actually for an eigenvalue $\Lambda_x(u)$ the ratio to the leading eigenvalue $\Lambda(u)$ of the transfer matrix

$$l(u) := \frac{\Lambda_x(u)}{\Lambda(u)}. \tag{A.10}$$

In the thermodynamic limit this function satisfies the functional equation (derived from two times (A.1) for $\Lambda(u)$ and for $\Lambda_x(u)$)

$$l(u)l(u-1) = 1. \tag{A.11}$$

This is solved uniquely for a given set of zeros $z_m$ in the physical strip by tanh resp. tan function (for any distribution of inhomogeneity parameters $\theta_j$). Let us assume there are only two such zeros $z_1$ and $z_2$ , then

$$l(u) = \tan\left( \frac{\pi}{2}(u - z_1) + \frac{1}{2} \right)\left( \frac{\pi}{2}(u - z_2) + \frac{1}{2} \right). \tag{A.12}$$

The shift $+\frac{1}{2}$ is due to the convention (23). The logarithmic derivative and then inserting $u = \pm a$ and $z_m = i\bar{z}_m$ gives directly (41).

However, the method requires that there do not exist the zeros between the lines $\text{Re}(z) = 0$ and $\text{Re}(z) = -1$ at the ground state. For example, we can know that zeros of the ground state in ferromagnetic regime are mainly located in line $\text{Re}(z) = -\frac{1}{2}$[1] from Section 7 . This will lead to an error in the Fourier transform (A.7).

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
