# Peer review of "Exact physical quantities of a competing spin chain in the thermodynamic limit"

_SciPost Physics, doi:SciPost Phys. 15, 060 (2023)_

## Round 1 · Referee Report · Anonymous (Referee 1) · 2022-12-14

Strengths

1- The authors compute various properties of an integrable spin chain with several couplings within an approach based on the roots of the corresponding transfer matrix eigenvalues.

Weaknesses

1- The implicit assumptions used in the analysis of the Bethe equations restrict their use to the computation of properties in the thermodynamical limit (and therefore, in the case of the isotropic model, effectively parallel boundary fields)

2- Results are not discussed in the context of previous ones obtained using conventional Bethe ansatz methods.

Report

The authors apply the Bethe ansatz scheme introduced in Refs. [22,23] to study bulk and boundary properties of the Heisenberg spin chain with competing interactions constructed from an inhomogeneous six vertex model with open boundary conditions. The scheme is based on the discrete set of discrete inversion relations (2.17). In this formulation boundary conditions breaking the $U(1)$ symmetry of the bulk system appear not to lead to the complications when dealing with unparallel boundary fields in the conventional TBA approach.

For the derivation of the integral form (4.2) of the Bethe equations, however, the authors have implicitely assumed that the inversion relations hold for a continuous variable $u$ -- not just at the special points $u=\theta_j$. This is not correct: for example, in this approach the spectrum would depend only on the effective parameters $p$ and $\bar{q}$ appearing in the eigenvalues of the boundary matrices (which determine the absolute value of the boundary fields) but not on the relative orientation of the boundary fields leading to the breaking of the $U(1)$ symmetry! In fact, it is well known that equations (2.17) only hold up to terms vanishing as powers of $(u-\theta_j)$.

Ignoring this fact (2.17) turns into a functional equation for the transfer matrix eigenvalues. Imposing constraints on their analytical properties (e.g. by considering a particular root pattern $\{z_j\}$) one selects an eigenstate and the functional equation can be solved directly by Fourier methods. As has been observed in previous works this yields the correct results for the bulk and boundary contributions to the corresponding energy in the thermodynamic limit (i.e. with boundaries being infinitely separated). Corrections of order $1/L$ can not be obtained in this way.

Therefore the results obtained in Section 4 based on the configurations of roots of the transfer matrix eigenvalues are correct although straightforward generalizations of what is known for the homogeneous Heisenberg model ($a=0$), see e.g. Grisaru et al., J. Phys. A28 (1995) 1027-1046 and Kapustin & Skorik, J. Phys. A29 (1996) 1629-1638.

Similarly, the spectrum of bulk elementary excitations (Sect. 5) has been studied for the periodic staggered ($a\neq 0$) spin chain before in Frahm & Rödenbeck, Europhys. Lett. 33 (1996) 47-52.

In Sect. 6 the authors study the boundary excitations and observe a different behaviour of their energy around $p=0$ for the homogeneous ($a=0$) and the staggered ($a\neq0$) spin chains. Given the $p$-dependence of the boundary term (2.3) this not really surprising.

In summary, most of the physical quantities considered in the manuscript are either known or straightforward extensions of known results which could have been obtained without using the 'novel Bethe ansatz scheme'. Moreover, the tacit assumption underlying the integral Eqs. (4.2) rules out an application of the proposed scheme to studies of the finite size spectrum. This would be necessary to address the particularly interesting case of unparallel fields advertised in the abstract and the introduction. Given these limitations I see little potential for the proposed scheme beyond what has already been done using established Bethe ansatz methods.

Therefore, I can not recommend to accept this manuscript publication in SciPost Physics.

Requested changes

1- The authors should extend their discussion the effect of the inhomogeneities $\theta_j$ at the end of Section 2. Also, only in Fig.2 their choice underlying the numerical data is stated. I assume that $\theta_j\equiv 0$ in Figs. 3, 6 and 7 (i.e. alternating inhomogeneities $\pm a$) but that should be clearly stated in the figure captions.

2- Summation indices in (4.1) should be $l$ and $k$, not $j$.

3- Where in the derivation of (4.3) has $\sigma(\theta)=\delta(\theta)$ been used?

4- The boundary elementary excitation (Sect. 6) should described more clearly: it is unclear how the root configuration of the excitation displayed in Fig. 7(a) for parameters from regime III can be obtained from the ground state one similar to that in Fig. 3(a) by just changing the boundary string (i.e. what happens to the roots at $\pm \alpha$ and $\pm i\beta$)?

  • validity: good
  • significance: ok
  • originality: low
  • clarity: low
  • formatting: -
  • grammar: -

Author:  Yi Qiao  on 2023-04-11  [id 3569]

(in reply to Report 1 on 2022-12-14)

Please see the attachments.

Attachment:

Reply_to_Reviewer1.pdf

---

## Round 1 · Referee Report · Anonymous (Referee 2) · 2022-12-27

Strengths

The problem is interesting, see attached pdf.

Weaknesses

The applied methods are cumbersome and the results are not as explicit as they can be, see attached pdf.

Report

See attached pdf.

Requested changes

Many calculations can be extremely shortened.

Attachment

  • validity: good
  • significance: ok
  • originality: ok
  • clarity: low
  • formatting: acceptable
  • grammar: below threshold

Author:  Yi Qiao  on 2023-04-11  [id 3570]

(in reply to Report 2 on 2022-12-27)

Please see the attachments.

Attachment:

Reply_to_Reviewer2.pdf

---

## Round 2 · Referee Report · Anonymous (Referee 2) · 2023-4-22

Strengths

see report

Weaknesses

see report: necessary clarification of (7.4) and (7.5)

Report

I found the replies by the authors to both reports satisfactory. The
manuscript has been amended and has gained significantly.

The authors are correctly pointing out that the elegant calculation presented
in Appendix A is not applicable to the ferromagnetic regime, because there the
eigenvalue function has an extensive number of zeros between the lines Re(z) =
0 and Re(z) = −1 at the ground state. However, for the ferromagnetic regime
many calculations simplify drastically, because in the conventional Bethe
Ansatz no Bethe roots appear for the ground state. Of course in the case of
general boundary fields this argument does not hold for the ODBA equations,
but the authors focus on the bulk O(N) and O(1) terms and ignore O(1/N) terms.

I am convinced that (7.4) must simplify considerably to something like N times
(J_1+J_2). Also, the explicit result in (7.5) resembles the terms in (2.3) and
(2.4), but there are differences. At this point the physical intuition tells
us that the bulk interactions favour a highly degenerate ground state of fully
polarized spins (in arbitrary direction). The calculations become simple and
can be done by elementary means. However, a fully polarized state will "see"
the differently oriented boundary fields and the result of the boundary energy
should depend non-trivially on the parameter \xi. However \xi dropped out in
the authors' calculation or has been set to 0 from the beginning.

Provided the authors clarify the last issue, I recommend the manuscript for
publication in SciPost.

Requested changes

necessary clarification of (7.4) and (7.5)

  • validity: high
  • significance: good
  • originality: good
  • clarity: good
  • formatting: -
  • grammar: reasonable

Author:  Yi Qiao  on 2023-05-09  [id 3660]

(in reply to Report 1 on 2023-04-22)

Please see the attachment.

Attachment:

reply2.pdf

Anonymous on 2023-05-26  [id 3689]

(in reply to Yi Qiao on 2023-05-09 [id 3660])

Thank you very much for your answers/explanations. Now the physical situation is clear to me.

---

## Round 2 · Referee Report · Anonymous (Referee 1) · 2023-4-24

Report

The authors have addressed the issues raised by the referees. Using the solution of the functional equation (2.22) for states close to the antiferromagnetic vacuum in the thermodynamic limit by direct Fourier transform their calculation of bulk and boundary properties has become more transparent. Some of the final results have been simplified.

The manuscript should be accepted for publication in SciPost Physics.

---

## Round 2 · Author Response

Dear editor,

Thank you for arranging a timely review for our manuscript submitted to SciPost.

We have substantially revised our manuscript after reading the comments provided by the reviewers. Those comments are all valuable and very helpful for improving our paper. Replies to the referee comments are attached as PDF files.

If you have any questions about this paper, please don't hesitate to let me know.

Yours sincerely
Yi Qiao

---

## Round 2 · List of Changes

We have revised the manuscript according to the referee's suggestions, and list the revisions as follows. The page numbers and equation numbers refer to revised version, unless specify.

  1. Added the explain of the index $j$ ``where the index $j$ is the summation index in $H_{bulk}$ (2.2)" after Eq.(2.6).

  2. Replaced In the derivation, we have used the relation $\sigma(\theta)=\delta(\theta)$." withFrom now on, we use $\sigma(\theta)=\delta(\theta)$." after (4.3).

  3. Replaced the words charity" withchiral three spin" in the manuscript.

  4. Added ``for ${\bar{\theta}_j= 0|j=1,\cdots,2N}$" in the captions of Figs. 3, 5, 6 and 7.

  5. Replaced the summation indices $j$ with $l$ and $k$ in the first and last of Eq.(4.1), respectivly.

  6. Redrawn Fig.7(a) and incorporated it with the root configuration of the ground state. Corresponding descriptions have also been added in the context of Sect. 6.

  7. Added some discussions of previous results obtained using conventional Bethe ansatz methods in line 7 on page 10 and after Eq.(5.2).

  8. Added some arguments about the finite size after Eq.(2.24) in line 2.

  9. Added a simplified and explicit form of the elementary excitation energy in Eq.(5.2) and Eq.(6.2).

  10. Modified the caption of Fig.7(a) to ``(a) The distribution of $\bar{z}$-roots for ${\bar{\theta}_j= 0|j=1,\cdots,2N}$ with $2N=8$, $a=0.66i$, $p=0.1$, $\bar{q}=1.2$ and $\xi=1.2$. Here the blue asterisks represent the pattern of zero roots at the ground state and the red circles denote those at the excited state with boundary string $i(\frac{1}{2}-|p|)$".

  11. Added a new section 7 to present the surface energies in ferromagnetic regime.

  12. Added a new Appendix A to present a clear and simple way recommended by an anonymous referee.

  13. Added references:

[30] M. T. Grisaru, L. Mezincescu and R. I. Nepomechie, {\it J. Phys. A: Math. Gen.} {\bf 28} 1027 (1995).

[31] A. Kapustin and S. Skorik, {\it J. Phys. A: Math. Gen.} {\bf 29} 1629 (1996)

[32] H. Frahm and C. R{\"o}denbeck, {\it Europhys. Lett.} {\bf 33} 47-52 (1996).

Besides, some words and sentences have also been slightly improved.

---

## Editorial Decision

published